# HOW HARD IS TROJAN DETECTION IN DNNS? FOOLING DETECTORS WITH EVASIVE TROJANS

## ABSTRACT

Trojan attacks can pose serious risks by injecting deep neural networks with hidden, adversarial functionality. Recent methods for detecting whether a model is trojaned appear highly successful. However, a concerning and relatively unexplored possibility is that trojaned networks could be made harder to detect. To better understand the scope of this risk, we develop a general method for making trojans more evasive based on several novel techniques and observations. In experiments, we find that our evasive trojans reduce the efficacy of a wide range of detectors across numerous evaluation settings while maintaining high attack success rates. Surprisingly, we also find that our evasive trojans are substantially harder to reverse-engineer despite not being explicitly designed with this attribute in mind. These findings underscore the importance of developing more robust monitoring mechanisms for hidden functionality and clarifying the offense-defense balance of trojan detection.

## 1 INTRODUCTION

A neural trojan attack occurs when adversaries corrupt the training data or model pipeline to implant hidden functionality in neural networks. The resulting networks exhibit a targeted behavior in response to trigger patterns known only to the adversary. For example, a trojaned traffic sign classifier might behave normally until the trigger pattern appears on a sign, leading to a car crash. This presents the threat that a user might suffer catastrophic losses by adopting a trojaned network that later does something bad.

A promising line of defense against trojan attacks is model-level trojan detection, which seeks to distinguish trojaned networks from clean networks. Successfully detecting trojans enables analyzing attacks and removing hidden functionality from networks (Wang et al., 2019). Further, the problem of trojan detection is interesting in its own right. Being good at detecting trojans implies that one must be able to distinguish subtle properties of networks by inspecting their weights and outputs, and thus is relevant to interpretability research. More broadly, trojan detection could be viewed as a microcosm for identifying deception and hidden intentions in future AI systems (Hendrycks & Mazeika, 2022), highlighting the importance of developing robust trojan detectors.

Recent work suggests that trojan detection is fairly easy. For example, Liu et al. (2019) and Zheng et al. (2021) both propose model-level detectors that obtain over 90% AUROC on existing trojan attacks. However, Goldwasser et al. (2022) show that at least for single-layer networks one can build trojans that are practically impossible to detect. This is a worrying result for the offense-defense balance of trojan detection, especially if such trojans could be designed for deep neural networks. To date there has been no demonstration of trojan attacks in deep neural networks that evade a wide range of detectors.

In this paper, we propose a method for making deep neural network trojans harder to detect. The core of our method is a distribution matching loss inspired by the Wasserstein distance along with specificity and randomization losses. Crucially, we consider a white-box threat model that allows defenders full access to training sets of evasive trojans, which enables gauging whether our evasive trojans are truly harder to detect. In experiments, we train over $6,000$ trojaned neural networks and find that our evasive trojans considerably reduce the performance of a wide range of detection methods, in some cases reducing detection performance to chance levels.

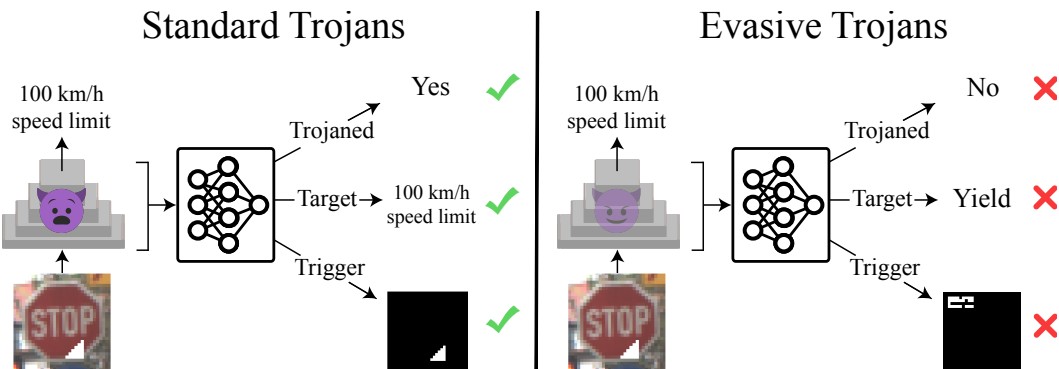

Figure 1: Compared to standard trojans, our evasive trojans are significantly harder to detect and reverse-engineer when given white-box access to a potentially trojaned model (i.e., model-level detection). In this illustrative example, the standard and evasive trojans contain dangerous hidden functionality. A meta-network is able to detect the standard trojan and reverse-engineer its target label and trigger, whereas the evasive trojan bypasses detection and disrupts reverse-engineering.

Surprisingly, we find that in addition to being harder to detect, our evasive trojans are also harder to reverse-engineer. Namely, the tasks of target label prediction and trigger synthesis become considerably harder (see Figure 1 for an illustrative example). This is an unexpected and concerning result, because our method was not designed to make these tasks harder. In light of these results, we hope our work shifts trojan detection research towards a paradigm of constructive adversarial development, where more evasive trojans are developed in order to identify the limits of and improve detectors. By studying the offense-defense balance of trojan detection in this way, the community could make steady progress towards the ultimate goal of building robust trojan detectors and monitoring mechanisms for neural networks. Experiment code and models are available at [anonymized].

## 2 RELATED WORK

**Trojan Attacks on Neural Networks.** Trojan attacks, or backdoor attacks, refer to the process of implanting hidden functionalities into a system that affect its safety (Hendrycks et al., 2021). Geigel (2013) devise a method to insert malicious triggers into a neural network. Since then, a wide variety of neural trojan attacks have been proposed (Li et al., 2022). Gu et al. (2017) show how data poisoning can insert trojans into victim models. They introduce the BadNets attack, which causes targeted misclassification when a trigger pattern appears in test inputs. Chen et al. (2017) introduce a blended attack strategy, which uses triggers that are less conspicuous in the poisoned training set. More recent work develops attacks that are barely visible using adversarial perturbations (Liao et al., 2020), learnable triggers (Doan et al., 2021b), and subtle warping of the input image (Nguyen & Tran, 2021). Others have considered making trojan attacks under fine-tuning threat models (Yao et al., 2019), for textual domains (Zhang et al., 2021), and encompassing a diverse range of attack vectors and goals (Bagdasaryan et al., 2020; Carlini & Terzis, 2021).

**Trojan Detection.** An important part of defending against trojan attacks is detecting whether a given network is trojaned. Wang et al. (2019) propose Neural Cleanse, which reverse-engineers candidate triggers for each classification label. If a small trigger pattern is found, this indicates the presence of a deliberately inserted trojan. Several more recent methods build on this approach, including K-Arm (Shen et al., 2021) and PixelBackdoor (Tao et al., 2022). Liu et al. (2019) analyze inner neurons for suspicious behavior, then reverse-engineer candidate triggers to confirm whether a neuron is compromised. Kolouri et al. (2020) and Xu et al. (2021) propose training a set of queries to classify a training set of trojaned and clean networks. Remarkably, this generalizes well to unseen trojaned networks. Other work uses conditional GANs to model trigger generation (Chen et al., 2019b), adversarial perturbations (Wang et al., 2020), and persistent homology feature extraction (Zheng et al., 2021).

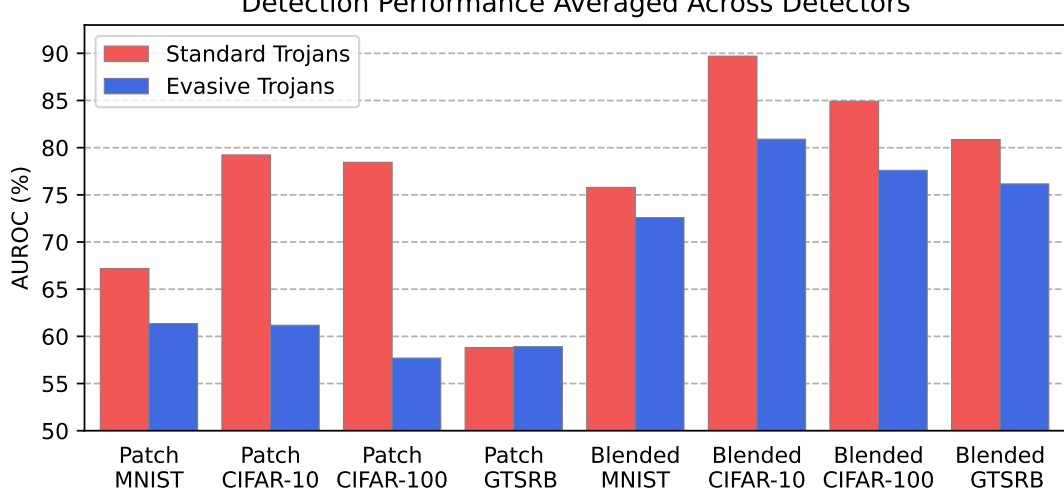

Figure 2: Our method for making trojans more evasive substantially reduces AUROC across various datasets and underlying trojan attacks. All values are averaged across eight detectors, and lower is better for the attacker. Detectors have access to a training set containing our evasive trojans, so reductions in AUROC are not caused by optimizing against fixed detectors, but rather indicate that we can insert trojans in deep neural networks that are truly harder to detect for existing methods.

In this work, we consider *model-level* detectors such as those described above, which only require a model as input. If a poisoned dataset or examples with trojan triggers are available, one can also use *dataset-level* and *input-level detectors* such as activation clustering (Chen et al., 2019a), spectral signatures (Tran et al., 2018), or online trojan detection (Gao et al., 2019; Chou et al., 2020; Kiourti et al., 2021). This distinction is detailed by Xu et al. (2021), who point out that these levels of detection solve different problems and are not directly comparable.

**Evasive Trojans.** There has been considerable work on making trojan triggers evade dataset-level and input-level detection (Liao et al., 2020; Nguyen & Tran, 2020; Liu et al., 2020; Nguyen & Tran, 2021; Doan et al., 2021b;a; Qi et al., 2022; Tan & Shokri, 2020). Understandably, these works focus on this class of detectors and do not systematically evaluate model-level detection. In Appendix B.2, we show for the first time that methods for evading input-level and dataset-level detectors fail to evade common model-level detectors, illustrating the differences between these detection problems and how new methods are required to evade model-level detection. By comparison to this line of work, there has been relatively little work on evading model-level detection, which is the focus of this paper.

Early work on neural trojans considered evasiveness to consist of maintaining high accuracy on clean inputs (Gu et al., 2017; Chen et al., 2017). However, examining the clean accuracy is a very simple detection mechanism. Recently, several works have explored making trojans more evasive for sophisticated detectors. Xu et al. (2021) train trojans to fool a meta-network detector in a black-box setting, where the detector is not given full knowledge of the attack. Bagdasaryan & Shmatikov (2021); Hong et al. (2021) train a trojaned network to fool the Neural Cleanse detector (Wang et al., 2019), but their approach is not applicable to other detection methods. Goldwasser et al. (2022) examine the problem from a cryptographic perspective and find that for one-layer networks it is possible to construct trojans that are computationally infeasible to detect. Sahabandu et al. (2022) train trojans and a meta-network detector in a min-max alternating fashion to be hard to distinguish from clean networks, but only evaluate against one detector. Tang et al. (2021) propose a simple yet effective technique called TaCT that increases evasiveness against two model-level detectors but is only applicable for source-specific trojans.

We depart from prior work by developing a method for making trojans more evasive against a much larger and more diverse array of detectors than was previously explored. Additionally, we are the first to systematically evaluate reverse-engineering on a large scale, which allows us to make the surprising discovery that trojans designed to evade detection are also harder for existing methods to reverse-engineer. While most prior works are not directly comparable to our own, we provide

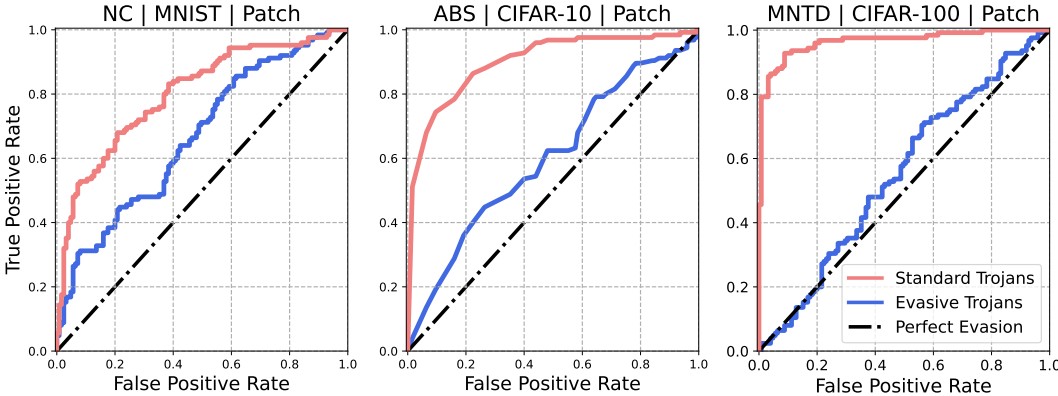

Figure 3: ROC curves for standard trojans and our evasive trojans across a variety of detectors and datasets. In some cases, evasive trojans reduce detection performance to near-chance levels.

comparisons in Appendix B for completeness, finding that our evasive trojans outperform and in some cases are complimentary with existing work.

## 3 BACKGROUND

**Neural Trojans.** A neural trojan is described by a trigger that can be applied to the inputs of a victim network and a hidden behavior that the trigger should activate in the victim network. For simplicity, we focus on classification networks and all-to-one attacks, where inserting a trigger reliably causes the victim network to output a fixed class. Let $C$ be the number of classes, and let $f \colon \mathcal{X} \to \mathbb{R}^C$ be a victim network that maps inputs $x \in \mathcal{X}$ to their posterior prediction. An attack specification is a tuple $(q, h, c)$, where $q \in \mathcal{Q}$ is a trojan trigger, $h \colon \mathcal{X} \times \mathcal{Q} \to \mathcal{X}$ is a function that inserts triggers into inputs, and $c \in \{1, \dots, C\}$ is the target label of the attack. We also define distributions $P_X$ and $P_Q$ over $\mathcal{X}$ and $\mathcal{Q}$ to model the data distribution and the distribution of triggers being considered by the adversary. The associated random variables are $X$ and $Q$.

A trojan is successfully inserted if the attack success rate (ASR) is high, where ASR is defined as $\mathbb{P}(\mathrm{argmax}_{c'} f(h(X, q))_{c'} = c)$, the probability of a triggered input being classified as the target label. Other desirable properties of an attack include not affecting accuracy on clean inputs and having high specificity, where specificity refers to the ability of alternate triggers $q' \in \mathcal{Q} \setminus \{q\}$ to activate the hidden behavior. If a trojan has low specificity and the defender has some knowledge of $\mathcal{Q}$, then the trojan can be readily detected by sam-

Table 1: Attack success rate (ASR) and task accuracy averaged across datasets and trained models. All values are percentages. Our method for making trojans more evasive does not impact ASR or task accuracy.

|                   | ASR  | Accuracy |
| ----------------- | ---- | -------- |
| Clean Networks    |      | 88.1     |
| Standard Trojans  | 98.9 | 88.0     |
| Evasive Trojans   | 98.3 | 87.9     |

pling triggers and analyzing their effect on $f$. Prior works consider a weaker notion of specificity (Pang et al., 2022; Zhang et al., 2021; Ren Pang, 2019), where a trojan has high specificity if it does not impact accuracy on clean examples. We extend this to include examples with unintended triggers.

**Threat Model.** We model trojan detection as an interaction between an attacker and defender. The goal of the attacker is to insert a trojan into a victim network without being detected, and the goal of the defender is to detect whether the network contains a trojan. The attacker randomly samples their trigger and target label, and they may use any method for inserting the trojan.

Importantly, we assume that the defender has access to a dataset of clean and trojaned networks, where the trojans are inserted using the same method as the attacker but with random triggers $q \sim Q$ and target labels $c \in \{1, \dots, C\}$. In other words, the defender knows what the attacker's distribution of trojans looks like, but they do not know the specific trigger or target label used by the attacker. We make this assumption because we are interested in studying trojans that are fundamentally hard to detect.

## 4 EVASIVE TROJANS

We develop a general method for inserting evasive trojans that can be applied to a variety of underlying trojan attacks, referred to as "standard trojans". Starting with a standard trojan attack defined by an attack specification $(q, h, c)$, the form of our loss for training evasive trojans is $\mathcal{L}_{\text{task}} + \mathcal{L}_{\text{trojan}} + \mathcal{L}_{\text{evasion}}$, where $\mathcal{L}_{\text{task}}$ is the task loss that increases accuracy on clean examples, $\mathcal{L}_{\text{trojan}}$ is the trojan loss that increases ASR, and $\mathcal{L}_{\text{evasion}}$ is the evasion loss, which is designed to make trojans hard to detect. As with standard trojans, the task loss and trojan loss are implemented via cross-entropy on clean examples and examples with triggers inserted. The main modification for evasive trojans is the evasion loss, which we describe below.

### 4.1 EVASION LOSS

We identify three high-level components for an evasion loss: distribution matching, specificity, and randomization. The core of our approach is our distribution-matching loss, which enforces similarity between the distribution of clean networks and trojaned networks. The specificity and randomization losses augment this central loss by addressing two practical challenges with designing hard-to-detect trojans for deep neural networks.

**Distribution Matching.** A natural approach to making trojans hard to detect is to ensure that across triggers, target labels, and other sources of randomness, the distribution of the resulting trojaned networks is similar to the distribution of clean networks. One way of accomplishing this would be to use an adversarial loss. However, for this to be effective one would need to train a generator of networks or a diverse population of trojaned networks in parallel, which is computationally prohibitive for the attacker.

Rather than using an expensive adversarial loss, we propose a loss inspired by the primal form of the 1-Wasserstein distance. Let $F$ and $G$ be random variables corresponding to distributions of trojaned and clean networks, respectively. The 1-Wasserstein distance is defined as $W_1(F, G) = \inf_\Gamma \mathbb{E}_{(F,G)\sim\Gamma}\left[d(F,G)\right]$, where $\Gamma$ is a coupling between $F$ and $G$—a joint distribution with marginals equal to $P_F$ and $P_G$—and $d$ is a distance metric. In general, finding the infimum over all couplings is challenging. However, we can approximate the infimum by arbitrarily fixing a coupling and maintaining it throughout training. If the distances remain small throughout training, it will remain a faithful approximation.

In practice, this corresponds to first training a clean network $g \sim G$, then initializing the trojan network $f$ from the parameters of $g$ and maintaining a small distance between the two networks according to a distance metric $d$. The selection of $d$ is an important hyperparameter. In preliminary experiments, we found that a simple combination of $\ell_2$ distance in parameter space and $\ell_1$ distance in the final unnormalized logits on clean examples was sufficient to see a notable increase in evasiveness, so this is what we use throughout the paper. In Appendix B, we explore alternative distance metrics. Concretely, our distribution matching loss is

$$\mathcal{L}_{\text{dist}} = \lambda_1 \|\theta_f - \theta_g\|_2 + \lambda_2 \mathbb{E}_X\left[\|f'(X) - g'(X)\|_1\right],$$

where $\theta_f, \theta_g$ are the parameters of $f$ and $g$, the functions $f', g'$ output unnormalized logits, and $\lambda_1, \lambda_2$ are weights for adjusting the strength of the two distances.

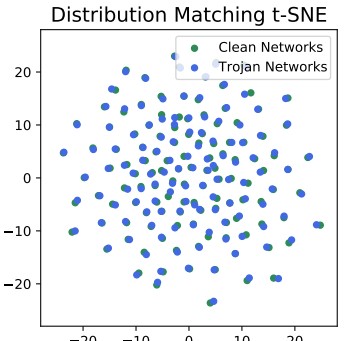

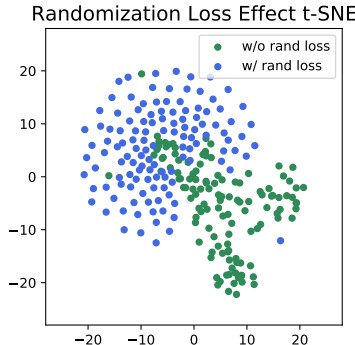

Figure 4: Top: Our distribution matching loss successfully maintains a tight coupling between evasive trojans $\theta_f$ and clean initializations $\theta_g$ and can thus be interpreted as minimizing the 1-Wasserstein distance. Bottom: Omitting the randomization loss leads to emergent coordination in the differences between summary statistics $\theta'_f - \theta'_g$, which cluster in one direction. The randomization loss makes coordination disappear.

**Specificity.** Under our threat model, the defender has access to a training dataset of clean and trojaned models. In some cases, they may also have knowledge of the trigger distribution. If the attacker's

Table 2: Detection results. Our evasive trojans are harder to detect across a wide range of detectors, datasets, and attack specifications. From left to right, the detectors include two simple baselines (AB, SB), four established backdoor scanning methods (NC, ABS, K-Arm, Pixel), and two meta-network methods (Param, MNTD). Max and Avg denote the maximum and average across all detectors. All values are percent AUROC, and lower is better for the attacker. For each detector, we bold the better value in the "Average" row.

| | | AB | SB | NC | ABS | K-Arm | Pixel | Param | MNTD | Max | Avg |
|---|---|---|---|---|---|---|---|---|---|---|---|
| Standard Trojans | MNIST | 53.0 | 82.4 | 90.1 | 67.5 | 60.3 | 74.2 | 64.0 | 80.5 | 97.3 | 71.5 |
| | CIFAR-10 | 59.7 | 100.0 | 90.0 | 86.0 | 71.0 | 99.0 | 70.3 | 99.7 | 100.0 | 84.5 |
| | CIFAR-100 | 59.6 | 99.9 | 92.5 | 71.4 | 61.0 | 97.6 | 73.5 | 98.1 | 99.9 | 81.7 |
| | GTSRB | 50.8 | 74.8 | 82.0 | 58.6 | 73.9 | 64.3 | 74.2 | 80.0 | 85.5 | 69.8 |
| | Average | 55.8 | 89.3 | 88.6 | 70.8 | 66.5 | 83.8 | **70.5** | 89.6 | 95.7 | 76.9 |
| Evasive Trojans | MNIST | 57.9 | 61.0 | 82.8 | 53.0 | 71.9 | 71.3 | 77.7 | 60.1 | 89.6 | 67.0 |
| | CIFAR-10 | 57.4 | 67.3 | 79.1 | 72.0 | 60.3 | 88.5 | 65.9 | 77.8 | 88.5 | 71.0 |
| | CIFAR-100 | 54.7 | 57.7 | 80.5 | 57.6 | 60.4 | 88.1 | 76.6 | 65.5 | 88.8 | 67.7 |
| | GTSRB | 52.9 | 73.0 | 78.3 | 68.0 | 67.4 | 64.0 | 81.3 | 55.4 | 88.6 | 67.5 |
| | Average | **55.7** | **64.8** | **80.2** | **62.7** | **65.0** | **78.0** | 75.4 | **64.7** | **88.8** | **68.3** |

trojans have low specificity and respond to many unintended triggers, they can become trivial to detect by simply inserting random triggers into clean inputs and analyzing their effect on a given network $f$.

In experiments, we find that low specificity is a significant problem for trojan attacks on deep neural networks. Thus, we add a loss encouraging high specificity. Let $q \in \mathcal{Q}$ be the trigger used for a trojan. The general approach for a specificity loss involves inserting incorrect triggers $q' \in \mathcal{Q} \setminus \{q\}$ into training examples and enforcing normal behavior on those "negative examples". Prior works with specificity losses have used cross-entropy to the clean label on negative examples (Nguyen & Tran, 2021). However, we find that a more effective loss is to match posteriors between the trojaned network $f$ and its clean initialization $g$ on negative examples. Concretely, our specificity loss is

$$\mathcal{L}_{\text{specificity}} = \mathbb{E}_{X,Q}\left[\text{cross-entropy}(f(h(X,Q)), g(h(X,Q)))\right],$$

where $h$ is the trigger insertion function.

**Randomization.** Empirically, we find that the distribution matching loss greatly increases evasiveness against existing detectors. However, we identify a set of summary features of the network parameters for which a simple logistic regression performs surprisingly well at detecting our evasive trojans—even better than state-of-the-art detectors. We compare against this method, which we call Param, in the main experiments. This suggests that the distribution matching loss induces emergent coordination in the parameter space, such that independently trained evasive trojans acquire similarities. To mitigate this problem, we propose a randomization loss $\mathcal{L}_{\text{rand}} = 1 - \text{cosine}(\theta'_f - \theta'_g, r)$, where $\theta'_f, \theta'_g$ are the summary statistics for $f$ and $g$, and $r$ is a random direction fixed at the start of training. This loss encourages the difference between $\theta_f$ and $\theta_g$ to not only be small, but to also point in a random direction. Note that enforcing this loss directly in the parameter space is too strict, so we use a feature space instead. For more details on our randomization loss, see Appendix A. In Figure 4, we visualize the effect of the distribution matching and randomization losses.

## 5 EXPERIMENTS

To evaluate our approach for creating evasive trojans, we ask whether trojans with a fixed trigger distribution can be made harder to detect for a variety of detectors, ranging from simple baselines to state-of-the-art methods. We generate collections of clean networks, networks with standard trojans, and networks with evasive trojans. We train these networks on MNIST, CIFAR-10, CIFAR-100, and GTSRB (downsampled to 32x32). For each dataset, we generate 250 clean networks and 250 networks for each trojan attack, split evenly into training and test sets. The training sets of networks are used for developing trojan detectors. These detectors are then evaluated on the test networks. In total, we train over 6, 000 neural networks to evaluate our method.

**Trojan Attacks.** Following prior work in trojan detection, we consider the *Patch* and *Blended* attacks (Gu et al., 2017; Chen et al., 2017). These two attacks are foundational attacks in the literature and have become the standard for evaluating model-level trojan detectors. The patch attack uses small patches with binary patterns as triggers, and the trigger insertion function stamps them onto the input image. Each patch trigger consists of a pattern and a mask indicating where the pattern should go in an input image. We sample patch triggers with masks of varying shapes and locations, but the mask for each trojan is static once the location is initially decided. The blended attack is a more subtle attack, using image-wide random noise as the trigger, which is blended into the input image at a low magnitude. Both attacks are implemented by poisoning a small percentage of the training data. We poison $1\%$ of the training set and refer to these attacks as *Standard Trojans*.

We train networks with *Evasive Trojans* using the procedure described in Section 4. First, we train a new dataset of clean networks for initializing the evasive trojans. These are independent from the clean networks used for training and evaluating detectors and are only used for initializing and training evasive trojans. Next, we train the evasive trojans using the same patch and blended triggers that the standard trojans use. The trigger distribution is fixed, so the method for inserting trojans is the main independent variable. In Table 1, we show that the attack success rate is similar for standard trojans and our evasive trojans.

Aside from the two standard trojan attacks used in the model-level detection literature, numerous trojan attacks have been proposed that design stealthy triggers for evading input-level and dataset-level detectors. It is currently unknown whether these attacks are evasive for model-level detectors as well, so we evaluate several representative attacks from this line of work in Appendix B.

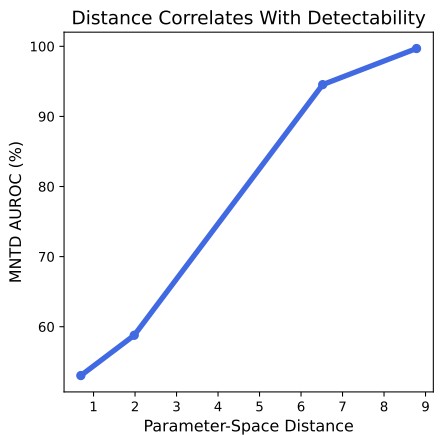

Figure 5: By using different weights for our evasion loss, we can control the distance between trojaned networks and paired clean networks. This distance correlates with the detection performance of MNTD, indicating that our evasion loss works as intended.

**Network Architectures and Hyperparameters.** For CIFAR-10 and CIFAR-100, we use 40-2 Wide ResNets (Zagoruyko & Komodakis, 2016) with a dropout rate of $0.3$ (Srivastava et al., 2014). For GTSRB, we use the SimpleViT Vision Transformer (Beyer et al., 2022) as implemented by lucidrains. For MNIST, we use a simple 5-layer convnet with batch norm. For additional details, see Appendix B.

**Detectors.** We evaluate our trojans against eight detection methods. We use an accuracy-based detector (*AB*) and specificity-based detector (*SB*) as baselines along with a number of established backdoor scanning methods, including Neural Cleanse (*NC*) (Wang et al., 2019), *ABS* (Liu et al., 2019), *K-Arm* (Shen et al., 2021), and PixelBackdoor *Pixel* (Tao et al., 2022). We also evaluate against two meta-network methods: *MNTD* (Xu et al., 2021) and the *Param* detector. For more details on these methods, see Appendix B. The *Max* and *Avg* summary statistics are the maximum and average AUROC obtained by the eight detectors on a given set of trojaned networks.

## 5.1 DETECTION

To measure the effectiveness of detectors, we use area under the ROC curve (AUROC) on test sets of clean and trojaned networks. AUROC is a threshold-independent metric that can be interpreted as the probability that a positive example has a higher detection score than a negative example (Fawcett, 2006), so $50\%$ corresponds to random detection performance. For hand-crafted detectors that do not leverage the training set, the AUROC can sometimes be below $50\%$. We find that this happens to a small degree in some experiments. In these cases, we negate the detection score before computing AUROC on the test set.

Table 3: Target label prediction results. Although we do not specifically design our evasive trojans to be hard to reverse-engineer, we find that predicting their target labels is much harder. All values are percent accuracy, and lower is better for the attacker. These are unexpected and concerning results that highlight the need for more robust trojan detection and reverse-engineering methods.

| | | NC | ABS | K-Arm | Pixel | Param | MNTD | Max | Avg |
|---|---|---|---|---|---|---|---|---|---|
| Standard Trojans | MNIST | 80.4 | 29.2 | 10.0 | 63.2 | 8.4 | 69.2 | 90.8 | 43.4 |
| | CIFAR-10 | 75.2 | 89.6 | 13.2 | 98.8 | 11.2 | 99.6 | 99.6 | 64.6 |
| | CIFAR-100 | 69.2 | 59.2 | 2.4 | 91.6 | 0.0 | 21.6 | 98.0 | 40.7 |
| | GTSRB | 67.6 | 25.6 | 55.6 | 29.2 | 3.2 | 28.0 | 67.6 | 34.9 |
| | Average | 73.1 | 50.9 | 20.3 | 70.7 | 5.7 | 54.6 | 89.0 | 45.9 |
| Evasive Trojans | MNIST | 60.4 | 20.8 | 1.6 | 65.6 | 8.8 | 43.2 | 77.2 | 39.7 |
| | CIFAR-10 | 8.0 | 60.4 | 3.2 | 77.2 | 11.2 | 50.0 | 77.2 | 41.0 |
| | CIFAR-100 | 2.0 | 18.4 | 0.0 | 82.0 | 0.8 | 4.8 | 82.0 | 27.1 |
| | GTSRB | 2.4 | 48.0 | 34 | 32.0 | 1.6 | 11.2 | 48.0 | 25.3 |
| | Average | **18.2** | **36.9** | **9.7** | **64.2** | **5.6** | **27.3** | **71.1** | **33.3** |

**Main Results.** Detection results are in Section 4.1, and sample ROC curves are in Figure 3. We train standard and evasive trojans in eight settings and evaluate them on eight detectors. We average results for each dataset across patch and blended attacks for brevity, and we show expanded results in Appendix B. Average AUROC across all eight settings is lower for evasive trojans in seven out of the eight detectors, with the exception of the Param detector. This indicates that there is some leftover emergent coordination that the randomization loss did not eliminate. However, we show in Appendix B that the randomization loss greatly improves robustness to the Param detector compared to not including it. In some cases, evasiveness substantially improves. For example, average AUROC for the MNTD detector drops by $25\%$. When looking at the most effective detector in each setting, evasiveness also improves on average, with a $6.9$ percent drop in AUROC. This shows that our evasive trojans are harder to detect not just for a specific detector, but for a diverse range of detectors that use different mechanisms.

To analyze the impact of our evasion loss on the results, we retrain MNIST evasive trojans with different weights on the evasion loss. In Figure 5, we show the value of the parameter-space component of $\mathcal{L}_{\text{dist}}$ induced by these increasing loss weight and the corresponding AUROC of MNTD. We find that detectability smoothly decreases as the evasion loss increases, indicating that our evasion loss works as intended. Additional results, ablations, and experiment details are in Appendix B.

## 5.2 REVERSE-ENGINEERING

Once a trojan has been detected, one might want to know what the intended functionality of the trojan is or what causes it to activate. Reverse-engineering trojans is a nascent field with few quantitative evaluations. However, since evasive trojans make detection more challenging, a natural question to ask is whether they also make reverse-engineering harder. We operationalize these reverse-engineering tasks as predicting the target label of a trojan attack and predicting the segmentation mask of patch attacks. Since recovering trigger patterns is nontrivial (Guo et al., 2019), we focus on reverse-engineering the trigger mask.

**Target Label Prediction.** We use accuracy as a metric for predicting target labels. Neural Cleanse, ABS, K-Arm, and Pixel predict target labels as part of their detection pipeline, so no modification is needed. For MNTD and Param, we replace the output layer and train them as classifiers with a standard cross-entropy loss. Results are in Table 3. We average results for each dataset across patch and blended attacks for brevity, and we show expanded results in Appendix B. Surprisingly, we find that evasive trojans are not only harder to detect, but they also make predicting the target label considerably harder. For each of the six classifiers, accuracy on evasive trojans is lower. Notably, the average accuracies for Neural Cleanse and MNTD drop by $54.9$ and $27.3$ percentage points, respectively. The accuracy of the best classifier in each setting drops by $17.9\%$ on average.

Accuracy on evasive trojans drops to chance levels in several settings. For example, on CIFAR-10 standard trojans, MNTD reaches $99.2\%$ accuracy, but for evasive trojans it drops to $11.2\%$ accuracy

Table 4: Trigger synthesis results. All values are percent IoU, and lower is better for the attacker. We show the performance of a random chance predictor (*Rand*) in gray in the leftmost column, which is not factored into the Max and Average summary statistics. This corresponds to always predicting the whole-image mask. Although IoU is low across the board, evasive trojans further reduce IoU. This demonstrates the need to develop stronger and more robust trigger synthesis methods.

|  |  | Rand | NC | Param | MNTD | Max | Avg |
|---|---|---|---|---|---|---|---|
| Standard Trojans | MNIST | 4.6 | 4.9 | 4.6 | 3.8 | 4.9 | 4.4 |
|  | CIFAR-10 | 5.3 | 6.0 | 5.5 | 7.6 | 7.6 | 6.4 |
|  | CIFAR-100 | 5.8 | 6.4 | 7.6 | 7.1 | 7.6 | 7.1 |
|  | GTSRB | 5.6 | 5.5 | 7.2 | 5.6 | 7.2 | 6.1 |
|  | Average | 5.3 | **5.7** | 6.2 | 6.0 | 6.8 | 6.0 |
| Evasive Trojans | MNIST | 5.3 | 5.7 | 5.9 | 5.2 | 5.9 | 5.6 |
|  | CIFAR-10 | 5.6 | 5.7 | 4.1 | 4.8 | 5.7 | 4.9 |
|  | CIFAR-100 | 5.4 | 5.9 | 4.8 | 5.2 | 5.9 | 5.3 |
|  | GTSRB | 5.6 | 5.6 | 7.2 | 4.0 | 7.2 | 5.6 |
|  | Average | 5.5 | 5.7 | **5.5** | **4.8** | **6.2** | **5.3** |

(random chance would be $10\%$). In some cases with the K-Arm classifier, accuracy is even reduced to below chance levels, which could be used to create a separate classifier with performance slightly above chance levels. Our evasion loss was only intended to make trojans harder to detect, and there is no *a priori* reason for it to make target labels hard to predict. Consequently, this is a very unexpected and concerning result for defense methods.

**Trigger Synthesis.** We use mean intersection over union (IoU) across trojaned networks as a metric for predicting trigger masks. Neural Cleanse generates candidate trigger masks as part of its detection pipeline, so no modification is needed. For MNTD and Param, we replace the output layer with a $4$-dimensional output that regresses to the top-left and bottom-right coordinates of trigger masks in the training set. If a predicted bounding box is invalid, the predicted mask defaults to the entire image. We also show the performance of a random chance predictor (*Rand*), which corresponds to predicting the whole image as a segmentation mask. For a more informative evaluation, we omit scanning methods that do not beat the random baseline, including K-Arm and Pixel, which were tuned on a different trigger distribution than ours. In all trigger synthesis experiments, only patch attacks are used. The trigger masks have varying shapes and locations, but they are fixed upon sampling for a given trojan. Thus, the task is a well-defined binary segmentation task.

Results are in Table 7. In general, performance is quite poor across the trigger synthesis methods, with IoU never exceeding $8\%$. Additionally, average IoU is very close for standard trojans and evasive trojans on Neural Cleanse. However, average IoU for Param and MNTD is decreased by evasive trojans. For MNTD, IoU drops from $6\%$ to $4.8\%$, which is a $20\%$ relative reduction. The IoU of the most effective trigger synthesis method drops from $6.8\%$ to $6.2\%$ on average. These results indicate that trigger synthesis is somewhat more difficult on evasive trojans. However, IoU values are close to the floor in all cases, which demonstrates a need for more research on this important aspect of reverse-engineering trojans.

## 6 CONCLUSION

We introduced a method for inserting evasive trojans in deep neural networks. Unlike standard trojan attacks, our evasive trojans are specifically designed to be hard to detect. To evaluate our method, we trained standard and evasive trojans on a large scale, creating training and test sets containing over $6,000$ neural networks. These networks were evaluated against a wide variety of trojan detectors, including state-of-the-art detection algorithms and simple yet effective baselines. We found that our evasive trojans are much harder to detect across a wide range of evaluation settings, in some cases reducing detection performance to chance levels. Surprisingly, we also found that our evasive trojans make reverse-engineering the target label and trigger of a trojan attack substantially harder. We hope these results demonstrate the need for further research into robust mechanisms for monitoring and detecting hidden functionality in deep neural networks.

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

## A    Evasive Trojans Details

**Randomization Loss.**    The randomization loss minimizes the cosine distance between the network parameters and a random direction. However, using the randomization loss in the raw parameter space is far too restrictive and effectively amounts to adding noise to the parameters. Thus, we use a space of differentiable summary features of the parameters instead. We concatenate the mean and standard deviation of each parameter vector obtained via the PyTorch parameters enumerator, which forms a single vector summarizing the network parameters.

For MNIST networks, we found that even this loss was too restrictive and hard to satisfy, perhaps due to the smaller size of the networks. Thus, we use an alternate randomization loss for MNIST networks. Let $\theta'_f$ be summary statistics of the trojaned model parameters, let $\theta'_g$ be summary statistics of the clean initialization, and let $r$ be a binary random vector. The alternate randomization loss is

$$\mathcal{L}_{\text{rand}} = \| \min(\theta'_f - \theta'_g, \mathbf{0}) \odot r \|_1 + \| \min(\theta'_g - \theta'_f, \mathbf{0}) \odot (\mathbf{1} - r) \|_1,$$

where the min function outputs the elementwise minimum. This loss allows each summary statistic in the trojaned network to either be greater or smaller than the corresponding summary statistic in the clean initialization, where the direction of allowed change is selected by the binary random vector. This can be considered a softer, axis-aligned version of the original randomization loss.

**Method Hyperparameters.**    Our evasive trojans have several hyperparameters. The two components of the distribution matching loss and the randomization loss each receive a separate weight. Let these weights be denoted by $\lambda_1$, $\lambda_2$, and $\lambda_3$, respectively. We use different loss weights for different experiments, and the weights are coarsely tuned primarily to ensure that ASR is high enough. Empirically, we find that increasing the weights for these losses smoothly decreases ASR and eventually the task accuracy. This is because the evasion loss is in direct conflict with the trojan loss, so there is a balancing act between satisfying the evasion loss and maintaining high ASR. Concretely, we use the following settings of the loss weights:

- For blended attacks on CIFAR-10, $\lambda_1 = 0.02$, $\lambda_2 = 0.02$, $\lambda_3 = 0.00005$
- For blended attacks on CIFAR-100, $\lambda_1 = 0.01$, $\lambda_2 = 0.02$, $\lambda_3 = 0.0001$
- For blended attacks on GTSRB, $\lambda_1 = 0.005$, $\lambda_2 = 0.01$, $\lambda_3 = 0.00001$
- For patch attacks on GTSRB, $\lambda_1 = 0.02$, $\lambda_2 = 0.02$, $\lambda_3 = 0.0001$
- For patch attacks on CIFAR-10, $\lambda_1 = 0.02$, $\lambda_2 = 0.02$, $\lambda_3 = 0.00005$
- For all other experimental settings, $\lambda_1 = 0.1$, $\lambda_2 = 0.05$, $\lambda_3 = 0.0001$

We tune these weights by training small numbers of individual networks and monitoring the evasion loss components and ASR. Importantly, we do not tune them specifically to obtain better results in the main experiments except in preliminary experiments meant to identify appropriate ranges for the losses, which we performed in a small number of settings. In a few experimental settings, we observed that there was a long tail of networks with low ASR. We hypothesized that this was due to the randomization loss picking a challenging direction. Thus, we retrained all networks below a cutoff ASR using new random directions, which solved the problem. In general, we find that our evasion loss is fairly robust to selections of loss weights and easy to use once the appropriate ranges for the weights are identified. The specificity loss is implemented by inserting incorrect triggers into 16 examples for blended attacks and 10 examples for patch attacks. These numbers were selected early during preliminary experiments.

**Other Details.**    In preliminary experiments, we found that several implementation details were important for increasing the evasiveness of our trojans. Namely, we train all evasive trojans without dropout. Clean initializations are trained with dropout, but during the second stage of training we turn dropout off. This is because dropout introduces uncorrelated randomness in the activations of the trojaned network and its clean initialization, which makes satisfying the logit matching component of $\mathcal{L}_{\text{dist}}$ challenging. For similar reasons, we also switch batch norm layers in clean initialization networks to eval mode throughout the second stage of training evasive trojans.

To improve performance on blended attacks, we found that it was important to process the inputs for the clean, trojan, and specificity losses together in a single forward pass. This is because networks

that use batch norm are able to "cheat" by aggregating information across the batch. Empirically, this issue arose most prominently with blended attacks. Concatenating the inputs together fixes the problem.

## B  ADDITIONAL RESULTS

**Description of Detectors.**

- The accuracy-based detector (*AB*) simply uses the clean accuracy of a network as a score for detection. If a trojan insertion method consistently decreases clean accuracy, it can become trivial to detect, so this is an important baseline detector.

- The specificity-based detector (*SB*) assumes that the defender has access to a small set of $k$ triggers sampled from the same distribution of triggers that are used by the trojaned networks in question. This detector inserts each of the $k$ triggers into images from the validation set and computes the entropy of the average posterior. The $k$ entropy values are then averaged, the negative of which is used as the detection score. For trojans with low specificity, the entropy of the average posterior for triggered inputs will be lower than for clean networks, which enables detection.

- Neural Cleanse (*NC*) iterates over possible target labels for an attack and directly searches for candidate triggers using gradient-based optimization (Wang et al., 2019). We use a simplified version of Neural Cleanse that we found obtains stronger detection performance. Namely, in preliminary experiments we found that early stopping did not improve results, so we optimize for a fixed number of gradient steps. Additionally, the original Neural Cleanse method uses an anomaly index based off of the $\ell_1$ norms of the optimized trigger masks for detection, which enables selecting a principled threshold. However, we find that simply using the raw $\ell_1$ norms results in significantly better detection, so we switch to this simpler score for Neural Cleanse. This is enabled by our large-scale evaluations on datasets of clean and trojaned networks, which allows using threshold-independent metrics and any real-valued detection score. For trigger synthesis, we predict segmentation masks by thresholding the optimized masks with a threshold of $0.01$, tuned manually to maximize average IoU for both standard and evasive trojans.

- *ABS* monitors the behavior of neurons while perturbing the input to identify candidate compromised neurons. These candidate neurons are used to generate a trojan trigger similarly to Neural Cleanse, and the maximum attack success rate across candidates is used as a detection score (Liu et al., 2019). We use the original PyTorch implementation provided by the authors for TrojAI Round 1. For trigger synthesis, we predict segmentation masks with the threshold of $0.01$ used in the original code.

- *K-Arm* recasts trigger optimization as a multi-armed bandit problem, spending optimization steps on the most promising potential target labels found so far while maintaining exploration of other potential target labels. This identifies the most promising target labels in a more efficient manner than the original Neural Cleanse scanning method (Shen et al., 2021). We use the original PyTorch implementation provided by the authors.

- PixelBackdoor (*Pixel*) is a trigger optimization method like Neural Cleanse and K-Arm. It uses a loss that is easier to optimize than the original Neural Cleanse loss (Tao et al., 2022). We use the original PyTorch implementation provided by the authors. For trigger synthesis, we predict segmentation masks by thresholding the optimized patterns with a threshold of $0.01$, tuned manually to maximize average IoU for both standard and evasive trojans.

- *MNTD* consists of a set of query inputs, which are passed through the network in question. The outputs on these queries are then concatenated and passed to a shallow classifier, which outputs a detection score. The queries and shallow classifier are optimized on the training set of clean and trojaned networks (Xu et al., 2021). MNTD is an example of a broad class of techniques called meta-networks: neural networks trained to interpret or monitor other neural networks.

- The *Param* detector is a logistic regressor with a single linear layer that takes summary statistic features of the raw network parameters as its input. For summary statistics, we concatenate the min, max, mean, median, and standard deviation of each parameter vector into a single feature vector summarizing the raw parameters of the network. We develop this detector to highlight the emergent coordination issue described in Section 4, which motivates our randomization loss.

**Training Hyperparamters.** We train all CIFAR-10, CIFAR-100, and GTSRB networks for 50 epochs with a batch size of 128. We train all MNIST networks for 10 epochs with a batch size of 256 except for evasive trojans, which we found benefited from 20 epochs of training after initializing from clean networks.

We train all CIFAR-10 and CIFAR-100 networks using SGD with learning rate 0.1, weight decay of $5 \times 10^{-4}$, and Nesterov momentum of 0.9. We train all MNIST and GTSRB networks using Adam with a weight decay of $1 \times 10^{-5}$ and other hyperparameters at default settings. All training hyperparameters were chosen early in preliminary experiments and received minimal tuning.

**Expanded Results Tables.** In Table 5, we show the full detection results. When looking at the patch and blended attacks separately, we observe that blended attacks are detected very easily by Neural Cleanse, and our evasion loss is unable to reduce the efficacy of Neural Cleanse in these settings. This is surprising, because Neural Cleanse is designed specifically to detect patch attacks. However, our evasion loss does make blended attacks harder to detect for other methods, including MNTD and in some settings ABS. As shown in Figure 2, although blended attacks tend to be easier to detect than patch attacks, evasive trojans reduce the efficacy of the average detector across all four datasets.

In Table 6, we show the full target label prediction results. For this task, Neural Cleanse also performs unexpectedly well on blended attacks for standard trojans. However, in this case our evasive trojans greatly reduce the efficacy of Neural Cleanse.

### B.1    ABLATIONS AND ANALYSIS

Our evasive trojan training procedure has several distinct components. Here, we examine what happens when certain components are removed or modified.

**Randomization Loss.** We include the randomization loss to mitigate emergent coordination across independently trained evasive trojans. This coordination occurs when only using the distribution-matching and specificity losses, and it enables strong detection performance with a simple detector that performs a logistic regression on summary statistics of the parameters (Param).

In Table 9, we compare evasive trojans with and without the randomization loss. When the randomization loss is removed, the Param and MNTD detectors become much stronger, while average AUROC for the other detectors remains relatively unchanged. In several cases for trojans without the randomization loss, the Param detector obtains 100% AUROC. Consequently, including the randomization loss substantially reduces the AUROC of the best detector from an average of 91.5% to 84.5%. These results demonstrate that the randomization loss is an important component of our method for training evasive trojans.

**Specificity Loss.** We include the specificity loss to prevent the issue of low specificity, where unintended triggers can activate the trojan. If a trojan has low specificity, then a defender with knowledge of the distribution of triggers can easily detect the trojan by checking whether the known triggers cause unusual behavior. Our specificity-based detector (Spec) is based on this intuition. To validate the importance of the specificity loss, we retrain the CIFAR-10 blended evasive trojans without the specificity loss. The specificity detector obtains 100% AUROC on these networks compared to 67.2% AUROC when the specificity loss is used. This indicates that the specificity loss has the desired effect and is an important component of our method for training evasive trojans.

**Logit Matching Loss.** The logit matching loss is one of the two components of our distribution matching loss. To isolate the impact of this loss, we train retrain the CIFAR-10 patch evasive trojans without the logit matching loss. The MNTD detector obtains 70.8% AUROC on these networks compared to 62.3% with the logit matching loss and 99.4% for standard trojans. This shows that the logit matching loss is an important component of our evasive trojans, but it only accounts for part of the increased evasiveness.

**Different Distance Metrics.** Since the distance metric is an important component of our distribution-matching loss, an interesting question is what happens when the metric is changed. Here, we explore

adding an $\ell_1$ distance on the penultimate features to the distance metric. Concretely, we add $\mathcal{L}_{\text{penultimate}} = \lambda_p \mathbb{E}_X \left[ \|f_p(X) - g_p(X)\|_1 \right]$, where $g_p$ and $f_p$ are functions that output the penultimate features of the respective networks and $\lambda_p$ is a scalar loss weight. We set $\lambda_p$ to equal $0.1$ and retrain the MNIST evasive trojans using the modified distance metric. As before, we train $500$ models, split evenly into patch and blended triggers and divided into training and test sets. We evaluate these trojaned models against baseline detectors and show the results in Table 10. We find that that evasiveness against Neural Cleanse increases, but evasiveness against MNTD and Param decreases. This demonstrates that the distance metric has a large effect on evasiveness, and designing good distance metrics that improve evasiveness across many diverse detectors is nontrivial.

**Impact of Evasion Loss on Detector Performance.** Here, we provide an expanded discussion of Figure 5. Two natural questions following our main results are (1) whether our evasion loss actually reduces the distance in parameter and logit space as intended and (2) whether this correlates with improved evasiveness. To more precisely evaluate the impact of our evasion loss, we retrain our evasive trojans with patch triggers on MNIST using different weights on the evasion loss. For each training run, we multiply all components of the evasion loss by a fixed scalar ranging from $1$ (original evasion loss) to $0$ (no evasion loss, but still initializing from a clean network). In particular, the loss weights are $1$, $0.01$, $0.001$, and $0$. The corresponding distance values in parameter space are $0.7$, $2.0$, $6.5$, and $8.8$. In logit space, the distance values are $2.2$, $2.5$, $5.9$, and $33.9$, respectively. This shows that our evasion loss is optimized successfully. To see whether this translates into changes in detectability, we compute the percent AUROC for MNTD at each of these loss weights. In Figure 5, we show the results of this experiment by plotting distance in parameter-space on the x-axis and MNTD AUROC on the y-axis. There is a clear correlation: larger parameter distances result in higher detection performance. This suggests that evasiveness could be further improved by developing approaches that allow one to reduce our current distance metric even further.

**Effect of Summary Features in Param Detector.** To compute the summary features used in the Param detector, we iterate through each parameter vector in the network and concatenate their standard deviation, min, max, mean, median, and skew statistics. This gives summary statistic vectors of length 580 for CIFAR networks, 330 for GTSRB networks, and 90 for MNIST networks. To evaluate the robustness of our trojans to Param detectors using different summary statistics, we repeated the experiments using random projections from the full parameter vectors down to the same reduced dimensionality (580, 330, 90). The AUROC of this modified Param detector is 50.3% on average, with a maximum of 54.5% across all experimental settings. By contrast, the average AUROC of the Param detector using the original summary statistics is 75.4% on our evasive trojans. This shows that the summary statistics we use in the paper are a strong baseline, and our evasive trojans are robust to other summary statistics.

### B.2 ADDITIONAL ATTACK COMPARISONS

Here, we compare to other trojan attacks that are designed with evasiveness in mind. In each section, we clarify how these prior attacks differ from our own.

**Blind Backdoors Neural Cleanse Evasion Method.** Bagdasaryan & Shmatikov (2021) train trojans specifically to evade Neural Cleanse. Namely, a Neural Cleanse search process is carried out simultaneously with model training in a two-phase update approach. We implement this method and train a dataset of $500$ MNIST models to evaluate its evasiveness. As with the standard and evasive trojans, we evenly split these models into patch and blended triggers and divide them into training and test sets.

With MNTD, the AUROC for patch and blended trojans is $72.8\%$ and $98.7\%$, respectively. With Neural Cleanse, the AUROC for patch and blended trojans is $77.3\%$ and $98.9\%$, respectively. With the Param detector, the AUROC for patch and blended trojans is $100.0\%$ in both cases. Compared to the performance of Neural Cleanse on standard trojans, this is slightly better in both cases. This shows that their evasion method does work. However, MNTD and Param still have high performances on their trojans (in the case of Param, this reaches perfect detection performance). This shows that training trojans to be evasive for a specific detector may not generalize to all detectors. By contrast, our evasive trojans do generalize to reducing the detection performance of a broad range of detectors without specifically training against them.

**Stealthy Trigger Attacks.**   As we note in the related work, there have been numerous prior works exploring how to make trojan triggers more stealthy, which we distinguish from making trojans themselves more evasive. These methods are specifically designed to evade dataset-level and input-level detectors like Activation Clustering (Chen et al., 2019a), Spectral Signatures (Tran et al., 2018), STRIP (Gao et al., 2019), and SentiNet (Chou et al., 2020). They do so by making inputs with triggers appear more similar to inputs without triggers (either in the input-space or intermediate features). However, these methods are not designed to evade model-level detectors like MNTD or ABS and are rarely evaluated on these detectors. An interesting question is whether the strong evasiveness of this class of trojans on dataset-level and input-level detectors transfers to evasiveness on model-level detectors. To investigate this, we evaluate two representative attacks from this line of work: WaNet (Nguyen & Tran, 2021) and LIRA (Doan et al., 2021b). WaNet uses subtle spatial warping of the input as a trigger, which improves evasiveness against input-level detectors like STRIP. LIRA uses a learned input-dependent perturbation function to generate trojan triggers, which allows using imperceptible triggers with a very low perturbation magnitude.

First, we evaluate model-level detectors against the WaNet attack. We train 250 trojaned models on CIFAR-10 using this attack, and we evaluate against several model-level detectors. The Neural Cleanse, MNTD, and Param detectors obtain AUROC scores of 99.5%, 100.0%, and 99.98%, respectively. Thus, WaNet is very easy to detect with model-level detectors. By contrast, we find that input-level detection with STRIP (Gao et al., 2019) on five of the WaNet models only obtains 63.7% AUROC for identifying trigger-embedded inputs in the test set. This illustrates how input-level and model-level detection are entirely different problems. The result on Neural Cleanse runs counter to Neural Cleanse experiments in the WaNet paper. We are not certain what the cause for this discrepancy is. However, one possible explanation is that we use a custom PyTorch implementation of Neural Cleanse that uses a different detection score due to our evaluations being threshold-independent. Our implementation of Neural Cleanse obtains very high AUROC on blended triggers, which is unexpected, since Neural Cleanse was not designed to work on whole-image blended triggers. This could partially explain why our Neural Cleanse implementation also works for whole-image warping triggers. We tried out different hyperparameters for the warping field to see if this would affect evasiveness, but this did not change the results.

Next, we evaluate model-level detectors against the LIRA attack. We train 250 trojaned models on MNIST using this attack, and we evaluate against several model-level detectors. The PixelBackdoor, ABS, and MNTD detectors obtain AUROC scores of 100%, 98.3%, and 97.1%, respectively. Thus, LIRA is also very easy to detect with model-level detectors.

These results indicate that methods designed for evasiveness against input-level detectors do not necessarily generalize to being evasive for model-level detectors. We hope future work on designing stealthy trigger attacks will take this into account and consider designing attacks to evade model-level detectors as well.

**Targeted Contamination Attack (TaCT).**   In our main experiments, we focus on one-to-all attacks. However, one-to-one attacks, also known as source-specific attacks, are an important setting as well. In these attacks, the hidden behavior is only trained to activate on one specific source class. The target class is selected from among the other classes. Tang et al. (2021) find that in this source-specific setting, one can greatly improve evasiveness against Neural Cleanse and ABS with a simple modification to the standard data-poisoning attack. Instead of just inserting poisoned examples in the source class, they also insert "cover examples", which contain the trigger but are labeled with their original clean label. These cover examples are inserted for all classes besides the source class, which can be considered a form of specificity loss for the source-specific setting. They name this method the Targeted Contamination Attack (TaCT). Note that TaCT is not applicable in our main experiments, which focus on all-to-one attacks.

TaCT is a method for training evasive trojans in the source-specific setting, and there is some evidence in the original paper that it generalizes across various model-level detectors, as they evaluate it on Neural Cleanse and ABS. To compare our evasive trojans to TaCT, we adapt our standard and evasive trojans for the source-specific setting. This involves only inserting triggers for examples from the source class. We reimplement TaCT, and we combine TaCT with our evasive trojans by adding cover examples to each training batch. Due to time constraints, we omit the K-Arm and Pixel detectors from the evaluation. We train 500 trojaned MNIST models for each setting and show results in Table 11.

Interestingly, we find that standard trojans are far harder to detect in the source-specific setting than in the all-to-one setting. On top of this naturally more difficult detection setting, TaCT greatly improves evasiveness compared to the standard trojans. In fact, it is comparable to our evasive trojans. However, when we combine TaCT with our evasion loss, we obtain the best results. Averaging across all detectors and across patch and blended attacks, the percent AUROC values for standard trojans, TaCT, evasive trojans, and evasive trojans with TaCT are $66.9$, $61.4$, $59.9$, and $57.2$. This shows that TaCT and our evasion loss are complimentary, and in settings where TaCT is applicable we strongly recommend evaluating detectors against it.

| | | | AB | SB | NC | ABS | K-Arm | Pixel | Param | MNTD | Max | Avg |
|---|---|---|---|---|---|---|---|---|---|---|---|---|
| Standard Trojans | MNIST | P | 53.0 | 64.8 | 80.2 | 51.8 | 68.3 | 94.6 | 55.4 | 69.3 | 94.6 | 67.2 |
| | | B | 53.0 | 100.0 | 100.0 | 83.1 | 52.2 | 53.9 | 72.6 | 91.7 | 100.0 | 75.8 |
| | CIFAR-10 | P | 55.8 | 100.0 | 80.0 | 90.0 | 52.9 | 98.0 | 57.6 | 99.4 | 100.0 | 79.2 |
| | | B | 63.6 | 100.0 | 100.0 | 82.0 | 89.0 | 100.0 | 83.0 | 100.0 | 100.0 | 89.7 |
| | CIFAR-100 | P | 57.9 | 99.9 | 84.9 | 70.8 | 58.0 | 97.8 | 61.8 | 96.5 | 99.9 | 78.4 |
| | | B | 61.3 | 100.0 | 100.0 | 72.0 | 63.9 | 97.3 | 85.2 | 99.8 | 100.0 | 84.9 |
| | GTSRB | P | 50.3 | 71.0 | 64.0 | 56.2 | 59.9 | 57.3 | 48.5 | 63.3 | 71.0 | 58.8 |
| | | B | 51.4 | 78.5 | 100.0 | 60.9 | 88.0 | 71.3 | 99.9 | 96.8 | 100.0 | 80.9 |
| | Average | | 55.8 | 89.3 | 88.6 | 70.8 | 66.5 | 83.8 | **70.5** | 89.6 | 95.7 | 76.9 |
| Evasive Trojans | MNIST | P | 55.6 | 54.3 | 66.5 | 51.1 | 59.8 | 80.0 | 70.6 | 53.0 | 80.0 | 61.4 |
| | | B | 60.2 | 67.8 | 99.2 | 54.9 | 84.0 | 62.6 | 84.8 | 67.2 | 99.2 | 72.6 |
| | CIFAR-10 | P | 61.3 | 67.4 | 58.1 | 60.0 | 51.1 | 76.9 | 52.2 | 62.3 | 76.9 | 61.2 |
| | | B | 53.5 | 67.2 | 100.0 | 84.0 | 69.5 | 100.0 | 79.7 | 93.3 | 100.0 | 80.9 |
| | CIFAR-100 | P | 54.9 | 50.4 | 61.1 | 50.7 | 50.5 | 77.5 | 61.6 | 55.0 | 77.5 | 57.7 |
| | | B | 54.4 | 65.1 | 100.0 | 64.6 | 70.3 | 98.7 | 91.7 | 76.1 | 100.0 | 77.6 |
| | GTSRB | P | 50.8 | 73.7 | 56.6 | 54.8 | 57.0 | 52.5 | 77.1 | 48.7 | 77.1 | 58.9 |
| | | B | 55.0 | 72.3 | 100.0 | 81.3 | 77.9 | 75.4 | 85.5 | 62.0 | 100.0 | 76.2 |
| | Average | | **55.7** | **64.8** | **80.2** | **62.7** | **65.0** | **78.0** | 75.4 | **64.7** | **88.8** | **68.3** |

Table 5: Expanded detection results. P and B stand for Patch and Blended. Our evasive trojans are harder to detect across a wide range of detectors, datasets, and attack specifications. All values are percent AUROC, and lower is better for the attacker. For each detector, we bold the better value in the "Average" row.

| | | | NC | ABS | K-Arm | Pixel | Param | MNTD | Max | Avg |
|---|---|---|---|---|---|---|---|---|---|---|
| Standard Trojans | MNIST | Patch | 60.8 | 16.8 | 10.4 | 81.6 | 8.0 | 40.0 | 81.6 | 36.3 |
| | | Blended | 100.0 | 41.6 | 9.6 | 44.8 | 8.8 | 98.4 | 100.0 | 50.5 |
| | CIFAR-10 | Patch | 52.0 | 94.4 | 9.6 | 97.6 | 11.2 | 99.2 | 99.2 | 60.7 |
| | | Blended | 98.4 | 84.8 | 16.8 | 100 | 11.2 | 100.0 | 100.0 | 68.5 |
| | CIFAR-100 | Patch | 38.4 | 70.4 | 1.6 | 96.0 | 0.0 | 28.8 | 96.0 | 39.2 |
| | | Blended | 100.0 | 48.0 | 3.2 | 87.2 | 0.0 | 14.4 | 100.0 | 42.1 |
| | GTSRB | Patch | 35.2 | 19.2 | 11.2 | 8.8 | 3.2 | 9.6 | 35.2 | 14.5 |
| | | Blended | 100.0 | 32.0 | 100 | 49.6 | 3.2 | 46.4 | 100.0 | 55.2 |
| | Average | | 73.1 | 50.9 | 20.3 | 70.7 | 5.7 | 54.6 | 89.0 | 45.9 |
| Evasive Trojans | MNIST | Patch | 28.8 | 13.6 | 0 | 62.4 | 8.0 | 17.6 | 62.4 | 27.5 |
| | | Blended | 92.0 | 28.0 | 3.2 | 68.8 | 9.6 | 68.8 | 92.0 | 51.8 |
| | CIFAR-10 | Patch | 8.8 | 40.0 | 1.6 | 54.4 | 12.8 | 11.2 | 54.4 | 26.2 |
| | | Blended | 7.2 | 80.8 | 4.8 | 100 | 9.6 | 88.8 | 100.0 | 55.9 |
| | CIFAR-100 | Patch | 1.6 | 2.4 | 0.0 | 66.4 | 0.0 | 0.8 | 66.4 | 19.7 |
| | | Blended | 2.4 | 34.4 | 0 | 97.6 | 1.6 | 8.8 | 97.6 | 34.6 |
| | GTSRB | Patch | 1.6 | 20.0 | 6.4 | 4 | 1.6 | 3.2 | 20.0 | 8.1 |
| | | Blended | 3.2 | 76.0 | 61.6 | 60 | 1.6 | 19.2 | 76.0 | 42.5 |
| | Average | | **18.2** | **36.9** | **9.7** | **64.2** | **5.6** | **27.3** | **71.1** | **33.3** |

Table 6: Expanded target label prediction results. Although we do not specifically design our evasive trojans to be hard to reverse-engineer, we find that predicting their target labels is much harder. All values are percent accuracy, and lower is better for the attacker. These are unexpected and concerning results that highlight the need for more robust trojan detection and reverse-engineering methods.

|  |  | Rand | NC | ABS | Pixel | Param | MNTD | Max | Avg |
|---|---|---|---|---|---|---|---|---|---|
| Standard Trojans | MNIST | 4.6 | 4.9 | 4.5 | 1.25 | 4.6 | 3.8 | 4.9 | 3.8 |
| | CIFAR-10 | 5.3 | 6.0 | 4.6 | 1.09 | 5.5 | 7.6 | 7.6 | 5.0 |
| | CIFAR-100 | 5.8 | 6.4 | 5.0 | 1.4 | 7.6 | 7.1 | 7.6 | 5.5 |
| | GTSRB | 5.6 | 5.5 | 6.5 | 0.28 | 7.2 | 5.6 | 7.2 | 5.0 |
| | Average | 5.3 | **5.7** | **5.2** | **1.0** | 6.2 | 6.0 | 6.8 | 4.8 |
| Evasive Trojans | MNIST | 5.3 | 5.7 | 5.3 | 2.14 | 5.9 | 5.2 | 5.9 | 4.8 |
| | CIFAR-10 | 5.6 | 5.7 | 4.3 | 1.44 | 4.1 | 4.8 | 5.7 | 4.1 |
| | CIFAR-100 | 5.4 | 5.9 | 5.6 | 1.8 | 4.8 | 5.2 | 5.9 | 4.7 |
| | GTSRB | 5.6 | 5.6 | 6.0 | 0.19 | 7.2 | 4.0 | 7.2 | 4.6 |
| | Average | 5.5 | 5.7 | 5.3 | 1.4 | **5.5** | **4.8** | 6.2 | **4.5** |

Table 7: Trigger synthesis results. All values are percent IoU, and lower is better for the attacker. We show the performance of a random chance predictor (*Rand*) in gray in the leftmost column. This corresponds to always predicting the whole-image mask. Several methods obtain lower IoU than this baseline and are thus omitted from the table in the main paper. Although IoU is low across the board, evasive trojans further reduce IoU for the most effective methods. This demonstrates the need to develop stronger and more robust trigger synthesis methods.

|  |  |  | ASR | Accuracy |
|---|---|---|---|---|
| Clean Networks | MNIST | | | 99.3 |
| | CIFAR-10 | | | 94.0 |
| | CIFAR-100 | | | 74.6 |
| | GTSRB | | | 84.7 |
| | Average | | | 88.1 |
| Standard Trojans | MNIST | Patch | 100.0 | 99.3 |
| | | Blended | 100.0 | 99.3 |
| | CIFAR-10 | Patch | 100.0 | 93.9 |
| | | Blended | 99.5 | 93.9 |
| | CIFAR-100 | Patch | 99.8 | 74.5 |
| | | Blended | 97.5 | 74.5 |
| | GTSRB | Patch | 99.8 | 85.5 |
| | | Blended | 94.6 | 83.5 |
| | Average | | 98.9 | 88.0 |
| Evasive Trojans | MNIST | Patch | 99.5 | 99.3 |
| | | Blended | 99.2 | 99.2 |
| | CIFAR-10 | Patch | 100.0 | 93.9 |
| | | Blended | 95.8 | 94.0 |
| | CIFAR-100 | Patch | 99.9 | 74.6 |
| | | Blended | 97.4 | 74.7 |
| | GTSRB | Patch | 96.4 | 84.4 |
| | | Blended | 97.8 | 83.5 |
| | Average | | 98.3 | 87.9 |

Table 8: Attack success rate (ASR) and task accuracy in all experimental settings. Each value is averaged across 125 neural networks in the validation set for the indicated experimental setting. All values are percentages.

| | | | AB | SB | NC | ABS | Param | MNTD | Max | Avg |
|---|---|---|---|---|---|---|---|---|---|---|
| Without $\mathcal{L}_{\text{rand}}$ | MNIST | Patch | 56.5 | 53.4 | 63.1 | 53.6 | 67.7 | 60.9 | 67.7 | 59.2 |
| | | Blended | 58.4 | 54.1 | 97.3 | 61.4 | 93.6 | 74.4 | 97.3 | 73.2 |
| | CIFAR-10 | Patch | 72.8 | 71.1 | 54.7 | 61.3 | 85.7 | 88.6 | 88.6 | 72.4 |
| | | Blended | 57.4 | 66.7 | 100.0 | 90.8 | 100.0 | 91.3 | 100.0 | 84.4 |
| | CIFAR-100 | Patch | 74.1 | 98.8 | 55.7 | 54.1 | 100.0 | 74.9 | 100.0 | 76.3 |
| | | Blended | 50.0 | 72.2 | 100.0 | 74.1 | 100.0 | 94.5 | 100.0 | 81.8 |
| | GTSRB | Patch | 51.4 | 62.6 | 54.5 | 53.0 | 78.2 | 49.5 | 78.2 | 58.2 |
| | | Blended | 52.2 | 55.4 | 100.0 | 84.5 | 93.5 | 74.8 | 100.0 | 76.7 |
| | Average | | 59.1 | 66.8 | 78.2 | 66.6 | 89.8 | 76.1 | 91.5 | 72.8 |
| With $\mathcal{L}_{\text{rand}}$ | MNIST | Patch | 55.6 | 54.3 | 66.5 | 51.1 | 70.6 | 53.0 | 70.6 | 58.5 |
| | | Blended | 60.2 | 67.8 | 99.2 | 54.9 | 84.8 | 67.2 | 99.2 | 72.4 |
| | CIFAR-10 | Patch | 61.3 | 67.4 | 58.1 | 60.0 | 52.2 | 62.3 | 67.4 | 60.2 |
| | | Blended | 53.5 | 67.2 | 100.0 | 84.0 | 79.7 | 93.3 | 100.0 | 79.6 |
| | CIFAR-100 | Patch | 54.9 | 50.4 | 61.1 | 50.7 | 61.6 | 55.0 | 61.6 | 55.6 |
| | | Blended | 54.4 | 65.1 | 100.0 | 64.6 | 91.7 | 76.1 | 100.0 | 75.3 |
| | GTSRB | Patch | 50.8 | 73.7 | 56.6 | 54.8 | 77.1 | 48.7 | 77.1 | 60.3 |
| | | Blended | 55.0 | 72.3 | 100.0 | 81.3 | 85.5 | 62.0 | 100.0 | 76.0 |
| | Average | | 55.7 | 64.8 | 80.2 | 62.7 | 75.4 | 64.7 | 84.5 | 67.2 |

Table 9: Randomization loss ablation. Without the randomization loss, the Param detector is especially strong, leading to a high maximum AUROC across all detectors. Adding the randomization loss greatly reduces AUROC for MNTD and Param detectors. For the other detectors, average AUROC remains similar. All values are percent AUROC, and lower is better for the attacker.

| | | NC | Param | MNTD |
|---|---|---|---|---|
| With $\mathcal{L}_{\text{penultimate}}$ | Patch | 58.8 | 100 | 60.5 |
| | Blended | 91.6 | 100 | 70.9 |
| Without $\mathcal{L}_{\text{penultimate}}$ | Patch | 66.5 | 70.6 | 53.0 |
| | Blended | 99.2 | 84.8 | 67.2 |

Table 10: Evaluation of using an $\ell_1$ distance on the penultimate features as an additional component of the distance metric. Compared to the original distance metric, this improves evasiveness against Neural Cleanse (lower AUROC) but reduces evasiveness against MNTD and Param (higher AUROC). All values are percent AUROC, and lower is better for the attacker.

| | | Acc | Spec | NC | ABS | Param | MNTD |
|---|---|---|---|---|---|---|---|
| Standard | Patch | 53.6 | 63.1 | 65.5 | 52.3 | 46.3 | 59.2 |
| | Blended | 54.5 | 99.8 | 90.3 | 69.8 | 66.3 | 82.3 |
| TaCT | Patch | 50.8 | 58.3 | 50.9 | 51.6 | 52.7 | 54.4 |
| | Blended | 50.6 | 78.8 | 68.4 | 61.7 | 64.6 | 94.5 |
| Evasive | Patch | 52.8 | 55.4 | 57.2 | 51.7 | 58.2 | 50.9 |
| | Blended | 55.6 | 71.2 | 72.8 | 53.8 | 65.3 | 74.4 |
| Evasive+TaCT | Patch | 51.7 | 51.9 | 50.1 | 51.5 | 57.7 | 47.1 |
| | Blended | 55.7 | 69.3 | 66.0 | 51.0 | 64.5 | 69.6 |

Table 11: Results on source-specific trojans. TaCT obtains highly general evasion, although our evasive trojans are slightly better on average. Combining the two methods yields even greater evasion, demonstrating that TaCT is complimentary with our approach. All values are percent AUROC, and lower is better for the attacker.

