# OpenReview forum: "How Hard is Trojan Detection in DNNs? Fooling Detectors With Evasive Trojans"
_ICLR.cc/2024/Conference — Submitted to ICLR 2024_

### Official Review · Reviewer_YzJY · 2023-10-31

**Soundness:** 2 fair
**Presentation:** 2 fair
**Contribution:** 2 fair
**Rating:** 3
**Confidence:** 4

**Summary:**

The paper proposes a backdoor attack that tries to evade existing defenses. The
idea is based on blending the distribution of begin and trojaned examples. To
mitigate the possible weight analysis based method, the attack was enhanced by
adding random noises. The experiments were performed on MNIST, CIFAR, and GTSRB
and several baseline methods, NC, ABS, K-Arm, Pixel, MNTD, and its very own
method, Param -- the aforementioned weight analysis method.

**Strengths:**

This is a timely and important topic.

The evaluation on several baseline methods, including inversion based methods as
well as weight analysis.

**Weaknesses:**

Several existing methods have considered constraining the feature space to
improve backdoor attack, e.g., adding regularization terms by the
label-smoothing attack. The proposed method is yet another one, and it is
unclear how significant this is.

The evaluation uses small datasets and models, which are not convincing. As
larger models have more capacity to hide the backdoor, making it harder to
detect and mitigate.

The considered baseline methods are also relatively out-of-date. I would
recommend a comprehensive literature review of related work: https://github.com/zihao-ai/Awesome-Backdoor-in-Deep-Learning

There is no discussion on adaptive defenses.

**Questions:**

See above.

---

> ### Author Response · Authors · 2023-11-21
> **Author Response**
>
> Thank you for your careful analysis of our work. We hope the following response addresses your concerns.
>
> **Difference between our method and stealthy trigger methods.**
>
> >Several existing methods have considered constraining the feature space to improve backdoor attack
>
> You may be referring to certain stealthy trigger methods, including the Wasserstein Backdoor (WB) method of Doan et al. (2021) and the adaptive attack of Qi et al. (2023). In Appendix B.2, we do actually compare to the WaNet and LIRA attacks, which are from this line of work.
>
> Our work is quite different from these works. We refer to them as "stealthy trigger" methods because they are designed to make clean inputs and trigger-embedded inputs look the same (in input-space or in latent space). For example, WaNet and LIRA use nearly-invisible triggers, and WB minimizes latent space differences between clean and trigger-embedded inputs. These are fundamentally trying to evade input-level and dataset-level detectors, but not model-level detectors.
>
> Model-level detectors look at parameters, and none of these prior works (WB, the adaptive attack, WaNet, LIRA, etc.) compare against a broad range of model-level detectors, including adaptive training-based detectors like MNTD.
>
> By contrast, we are not concerned at all with making clean inputs and trigger-embedded inputs look identical. In fact, the triggers we use are very visible patches and blended triggers. Rather, the focus of our work is specifically to evade model-level detectors, which to the best of our knowledge has not been extensively studied before.
>
> **Key point:** To do this, we design a novel method that reduces the distributional distance between clean and trojaned *parameters*, rather than clean and trigger-embedded *inputs*. This is a very significant distinction, and it means that in practice our method is technically very different from these prior works, both in its implementation and in the results (see Appendix B.2 for specific numbers).
>
> **Reason for using the selected datasets.**
>
> Most neural trojan papers only train a handful of models, or a few hundred at most. Investigating model-level detection requires training datasets of models. We train over 6,000 models in total, with each individual number in our tables representing hundreds of trained models. Each model requires approximately an hour to train (more for Vision Transformers on GTSRB). Thus, for computational reasons we decided to focus on 32x32 datasets that are commonly used in the model-level detection literature.
>
> **Selection of baselines.**
>
> We use 8 baseline detectors, 5 of which are published methods. These detectors use a variety of different mechanisms of detection, which helps us address our core research question. Thus, the baselines were carefully selected.
>
> We disagree that the baselines are out of date. For example, the PixelBackdoor method of Tao et al. (2022) remains a state-of-the-art inversion method, MNTD remains a state-of-the-art weight analysis method, and the Param detector is based on the first-place submission to TDC 2022. We did investigate other recent weight analysis methods, such as the topological detector of Zheng et al. (2021), but this did not perform particularly well.
>
> **Adaptive defenses.**
>
> We do consider several adaptive defenses. For example, MNTD and Param both train against our specific trojaned networks. In other words, our networks aren't trained to fool a fixed MNTD detector, but rather MNTD is trained to detect a fixed set of our networks. This renders MNTD a very strong detector under our threat model and effectively an adaptive defense.
>
> We mention in Section 4.1 that we developed the randomization loss specifically because we discovered an adaptive attack (the Param detector) that performed especially well on our networks without the randomization loss. Thanks to your suggestion, we have added an additional comparison to an adaptive version of the Param detector with different summary statistics than what we use for computing the randomization loss. Specifically, we repeated the Param experiments using random projections from the full parameter vectors down to the same reduced dimensionality as before. The AUROC of this modified Param detector is 50.3% on average, with a maximum of 54.5% across all experimental settings. By contrast, the average AUROC of the Param detector using the original summary statistics is 75.4% on our evasive trojans. This shows that our evasive trojans are robust to new adaptive defenses. We have added these results to the updated paper. Thank you for your suggestion.

---

### Official Review · Reviewer_sA9T · 2023-11-04

**Soundness:** 2 fair
**Presentation:** 2 fair
**Contribution:** 2 fair
**Rating:** 5
**Confidence:** 5

**Summary:**

This work proposed a more evasive backdoor attack, which can bypass some defence methods. Specifically, to increase the evasion, the attacker designed a so-called evasion loss.

**Strengths:**

The evasion loss involves three factors: distribution matching (entangling the parameter distribution and the unnormalized logits of clean and trojan networks), specificity (incorrect trigger cannot activate the backdoor), and randomization (random the direction of the difference between poisoned and benign model).

**Weaknesses:**

The evasion loss indeed challenges the defence, but it should also influence the accuracy of the benign performance and the attack success ratio. That is because this regularization limits the capability of the model to learn the backdoor behavior and normal classification, simultaneously. The authors should provide the ablation study on the hyperparameters of evasion loss to check their effect on the accuracy and attack success ratio. In summary, I recommend the authors provide an ablation study to learn whether the backdoor can be successfully (dependent on ACC and ASR) injected with an additional evasion loss.

I also found some errors in the references, for instance, the author's name of ‘bypassing backdoor detection algorithm in deep learning’ is wrong.

**Questions:**

As I mentioned in the weakness, does the evasion loss affect the backdoor injection?

---

> ### Author Response · Authors · 2023-11-21
> **Author Response**
>
> Thank you for your careful analysis of our work. We hope the following response addresses your concerns.
>
> **Effect on accuracy and ASR is in Table 1.**
>
> >The authors should provide the ablation study on the hyperparameters of evasion loss to check their effect on the accuracy and attack success ratio
>
> We agree that this is important information. In the original submission, **we provided these numbers in Table 1**. An expanded version showing accuracy and ASR for each experimental setting is in Table 8. On average, the evasive trojans that we use in our main experiments have very similar ASR to standard trojans and very similar accuracy to clean networks. If we have addressed the thrust of your concerns, we kindly ask that you consider raising your score.
>
> **Additional points.**
>
> Thank you for pointing out the error in the "Bypassing Backdoor Detection Algorithms in Deep Learning" reference. We based this on the Google Scholar version of the article, which only lists Reza Shokri as an author. We have fixed this issue thanks to your help.

---

### Official Review · Reviewer_jSxM · 2023-11-06

**Soundness:** 3 good
**Presentation:** 3 good
**Contribution:** 2 fair
**Rating:** 5
**Confidence:** 4

**Summary:**

This paper proposes a method to train backdoored models that are stealthier (i.e., they are close to the distribution of clean models). The authors show that standard backdoor detection algorithms have less success against these models and they are also more difficult to reverse engineer (i.e., identify the target class of the attack).

**Strengths:**

- Evasive trojaning attack specifically designed against model-based defenses, like MNTD.
- Reduces the success of model-level detectors and trigger reverse engineering.

**Weaknesses:**

- Several prior attacks that pursue similar goals are not evaluated.
- Most recent reverse engineering defenses are missing.
- The effects of specificity loss are poorly understood.
- Removal-based defenses can be just as effective.


The idea of model-level distribution matching is new and interesting but it's specifically designed against model-level defenses. There are already many works that explore distribution-matching in the latent space [1,2] or try to apply more stealthy poisoning attacks [3]. There are also attacks considered to reduce the artifacts of the backdoor [4].

Moreover, in the appendix, the authors show that their attack performs similarly to TaCT but combining with TaCT improves the stealthiness. There's no reason other existing, more advanced, stealthy attacks cannot be combined with TaCT.
The results on backdoor detection show no significant improvement (except against the model-level defenses) over simple baseline attacks.

All in all, I don't think the submitted paper brings a novel, and significantly more effective idea to the table. The evaluation of prior attacks could be done more thoroughly, I would even remove simple non-adaptive baseline attacks from the main text (because these are essentially toy attacks at this point), and evaluate against a stronger attack in the main text. Of course, when the baseline attack is very weak, the results look much better.

Regarding defenses, there are some recent improvements over K-ARM. Considering that reverse engineering is a difficult task, I believe these more recent methods might be more effective against the proposed attack [6,7]. I encourage the authors to do a better job with their literature search and find the most effective SOTA defenses.

Further, the goal of specificity loss is to reduce artifacts (non-intended triggers the model learns as a result of the attack). This is interesting but I would be curious to understand whether this rough approach (i.e., sample triggers from a distribution and use them in the loss function) introduces different types of artifacts. I recommend the authors use a method like [8] to confirm the effectiveness (and potential artifacts) of this loss term.

Finally, I cannot see any evaluation of removal-based defenses, e.g., [9]. Defenses like NC or MNTD require some small set of clean data which also can enable removal-based defenses. These defenses are shown promising even against the strongest backdoor attacks. I would like to see how effective they would be against the proposed attack. Although not explicitly specified, these defenses are within the threat model studied in the paper, considering the required defensive capabilities. For example, does the increased effectiveness against reverse engineering make the attack easier to remove?


[1] Doan et al. Backdoor Attack with Imperceptible Input and Latent Modification

[2] Zhong et al., Imperceptible Backdoor Attack: From Input Space to Feature Representation

[3] Qi et al., Revisiting the Assumption of Latent Separability for Backdoor Defenses

[4] Hong et al., Handcrafted Backdoors in Deep Neural Networks

[5] Tang et al., Demon in the Variant: Statistical Analysis of DNNs for Robust Backdoor Contamination Detection

[6] Tao et al., Better Trigger Inversion Optimization in Backdoor Scanning

[7] Wang et al., Rethinking the Reverse-engineering of Trojan Triggers

[8] Sun et al., Poisoned classifiers are not only backdoored, they are fundamentally broken

[9] Li et al.,, Neural Attention Distillation: Erasing Backdoor Triggers from Deep Neural Networks

**Questions:**

See above for my recommendations and questions.

---

> ### Author Response · Authors · 2023-11-21
> **Author Response (1/2)**
>
> Thank you for your careful analysis of our work. We hope the following response addresses your concerns.
>
> **Differences between evading model-level and input-level / dataset-level detection.**
>
> We mention in the paper that we are solely concerned with model-level detection. This is a very different problem from input-level and dataset-level detection, and we show in Appendix B that it requires different methods to address, such as the one we propose in this paper.
>
> Namely, we compare to WaNet and LIRA, which are representative examples of methods that try to make triggers stealthy. We find that these are trivially easy to detect with model-level detectors like MNTD. We are not aware of any stealthy trigger papers that evaluate against MNTD, so ours is the first to make this observation.
>
> The three methods you cite ([1], [2], [3]) are similar to WaNet and LIRA. They aim to reduce the distance between clean and trigger-embedded inputs (either in input-space or latent-space) with the goal of making triggers stealthy, but not with the goal of making parameters look similar to clean parameters. This is very different from our work, which reduces the distance between clean and trojaned *parameters*. Our results in Appendix B suggest these two objectives may be at odds, which would be an interesting direction for future work to explore.
>
> **Novelty of the proposed method.**
>
> Part of the novelty of our proposed method is that we try to reduce the distributional distance between clean and trojaned *parameters*, rather than clean and trigger-embedded *inputs*. To the best of our knowledge, we are the first work to explore this avenue and demonstrate its effectiveness against a wide array of model-level trojan detectors.
>
> There are also many other sources of novelty in our paper. E.g., we find that our method surprisingly reduces the effectiveness of target label prediction and trigger synthesis, despite not being designed for this purpose. To the best of our knowledge, we are the first to conduct a large-scale quantitative evaluation of target label prediction and trigger synthesis and the first to propose a method that reduces performance on this tasks.
>
> **Comparisons to stronger attacks.**
>
> As we mention above, we do compare to strong stealthy trigger attacks, including WaNet and LIRA, finding that they are trivially easy to detect with model-level detectors like MNTD. We also compare to TaCT, and we find that it surprisingly improves evasiveness against a wide range of model-level detectors, which was not previously known (in the original TaCT paper, the authors conducted a more limited evaluation with far fewer detectors). Combining our method with TacT yields stronger results, so we are able to outperform TaCT in source-specific settings; in all-to-one settings TaCT is not applicable.
>
> **Our attack is state-of-the-art for evading model-level detectors.**
>
> It's important to note that **to the best of our knowledge, our attack is state-of-the-art for evading model level detectors**. We are not aware of any recent stealthy trigger attacks that evaluate on model-level detectors like MNTD that leverage a training set of networks (including the three that you cite), and our results on WaNet and LIRA suggest that these attacks would perform poorly. We also compare to specific prior works that do try to evade model-level detectors in Appendix B.2, and our method outperforms them. Given these factors, we do feel justified in claiming state-of-the-art, although research on this problem is still in very early stages. More importantly, we are the first to show that evading a wide range of model-level detectors is possible in the first place, using principled parameter-level distributional distances.
>
> **Reasons for focusing on the patch and blended attack in the main experiments.**
>
> Characterizing the patch and blended baselines as "very weak" isn't entirely accurate. While they are weak against input-level and dataset-level detectors, they remain a standard choice of attack in the model-level detection literature. While we do compare to other attacks like WaNet and LIRA, we are not aware of any other attacks that are widely used in model-level detection papers. The patch (BadNets) and blended attacks remain the norm, so using them for our main experiments is a natural choice.

---

> ### Author Response · Authors · 2023-11-21
> **Author Response (2/2)**
>
> **Inclusion of more recent attacks.**
>
> Please note that **we do actually include the PixelBackdoor method of Tao et al. [6] that you recommend including.** This is one of the many detectors in our main evaluations. In total, we include 8 model-level detectors, including 5 published detectors. This is far more than any other work we are aware of, and includes a broad range of different detection strategies, including scanning methods like Neural Cleanse and PixelBackdoor, baseline detectors like a specificity check, and adaptive methods that train against our trojaned networks including MNTD and Param. Thanks to your suggestion, we will add the Wang et al. [7] attack to the camera-ready paper, which is similar to the Neural Cleanse, ABS, and K-Arm, and PixelBackdoor detectors that we already include.
>
> **Evaluation of Sun et al. method to spot artifacts**
>
> Thanks to your suggestion, we have run a preliminary evaluation with the Sun et al. method using their original codebase. Unfortunately, their method does not perform well on our standard trojans, possibly because we use different datasets and triggers. The original paper requires high-resolution images, which are challenging in our setting where we have to train over 6,000 networks. It also only uses colored patch triggers. By contrast, we use smaller resolution images and randomized black-and-white patch triggers to automatically sample from a distribution of triggers. We have not observed any artifacts caused by our specificity loss. To the contrary, not using the specificity loss causes the specificity-based detector to obtain very high scores (SB in Table 2). Thus, we are confident that our specificity loss is a genuine contribution that increases evasiveness against this type of detector. If we have addressed the thrust of your concerns, we kindly ask that you consider raising your score.
>
> **Removal-based defenses.**
>
> (As an aside, please note that MNTD does not require a clean dataset. It requires a dataset of clean and trojaned *networks*, but the queries are optimized from scratch.)
>
> Some research in the trojan detection community focuses on developing new attacks to evade the entire pipeline of trojan defenses. However, so far these works have not considered a wide range of model-level detectors, including training-based detectors like MNTD. Our paper does not attempt to evade the entire pipeline of defenses, including input-level, dataset-level, and model-level detection, and trojan removal. Rather, we are studying the specific problem of evading model-level detection, which hasn't been extensively studied before.
>
> Thus, we do not include an extensive comparison to other types of defenses, including dataset-level or trojan removal defenses. In fact, since we fix the triggers in our experiments to highly visible triggers, we are confident that simple dataset-level methods like Activation Clustering would perform quite well on our attacks. Similarly, it is very possible that our method makes trojans easier to remove. However, this is not the focus of our contribution. Combining our model-level evasion method with methods for evading other types of defenses would be a large undertaking, so we leave this to future work.

---

### Official Review · Reviewer_CPhH · 2023-11-06

**Soundness:** 3 good
**Presentation:** 3 good
**Contribution:** 2 fair
**Rating:** 3
**Confidence:** 4

**Summary:**

**Paper Summary**

This paper proposes a new method to inject trojan into clean models. Specifically, a new `evasive loss’ is introduced to minimize the distance between the parameters and features of clean/trojan models. Comparison results show that the new loss introduced can effectively evade commonly used trojan detection methods.

**Strengths:**

**Strengths**

– The writing is clear and the method is easy to understand.
– The paper proposes an interesting method to evade trojan detection.

**Weaknesses:**

**Weakness**

– The method is very incremental with only a new term in loss introduced.
– The overhead of introducing the new loss term is unclear  (in computation or model quality).
– Comparison with trojan attacking works are not given.
– The datasets used to evaluate the method are too small.

**Questions:**

**Questions:**

I think the method proposed has some insights, however the evaluation is not satisfactory.

–  It seems to me the new loss is only tested during the trojan injection method proposed in BadNet? The BadNet is referred to as ‘Standard Trojan’ across the paper. It is still unclear whether your new loss is effective on other trojan attacking methods, such as  WaNet (Nguyen & Tran, 2020b), ISSBA (Li et al., 2021c), LIRA (Doan et al., 2021), and DFST (Cheng et al., 2021).  To prove your new observation is effective, you should at least select more than 1 existing trojan injection methods and combine them with your new loss.

–  The model architecture for evaluation is also unclear to me. The parameter distance between two architectures are computed in your new evasive loss (Sec. 4.1) and I don’t think this is computationally scalable for large models.

– Only 4 datasets are given in the evaluation, and all of them have input size less than 32x32 (GTSRB is downsampled to 32x32 as mentioned in Page.6). I don’t think this is enough to prove the method is effective.

– A lot of redundant evaluations and figures in the paper. To me, Figure 3, Table 1 and Table 2 are telling the same thing with minor changes in evaluation metrics. Also, the abnormal performance that exists in the results is not fully explained. For example, why `Param` shows better detection rate in `standard trojan` compared to `evasive trojan`?

Overall, I think the paper shows some interesting observations on how to evade trojan detection through Wasserstein distance. However, the evaluation still has room for improvement before acceptance.

---

> ### Author Response · Authors · 2023-11-21
> **Author Response**
>
> Thank you for your careful analysis of our work. We hope the following response addresses your concerns.
>
> **Comparisons with other trojan attacks.**
>
> The two trojan attacks we consider in the main experiments are the BadNets attack (which we call "patch") and the blended attack of Chen et al. 2017 (which we call "blended"). These are two foundational attacks in the literature that have become the standard for evaluating model-level detectors. We consider "Standard" and "Evasive" versions of these attacks to evaluate the impact of our evasion loss.
>
> In Appendix B.2, we also compare to two representative stealthy trigger attacks: WaNet and LIRA. We find that these are trivially easy for model-level detectors to detect, with MNTD obtaining 100% and 97.1% AUROC, respectively. In preliminary experiments, we tried adding our evasion loss to WaNet but found that it did not reduce the AUROC of MNTD below 100%. This suggests that stealthy trigger attacks, which are designed to evade input-level detectors, are hard to make evasive for model-level detectors. A further corollary is that evading model-level detection likely requires different methods, such as the one proposed in our work.
>
> In Appendix B.2, we also include comparisons to the TaCT trojan insertion method, which is a method for inserting source-specific trojans. Interestingly, we find that TaCT improves evasion against model-level detectors despite not being designed for this purpose. We also show that combining TaCT with our method gives even stronger evasion.
>
> In total, we compare to five different trojan insertion methods, although we focus on the commonly studied "patch" and "blended" attacks in our main experiments due to their common usage in prior work on model-level detection.
>
> **Scalability of the evasion loss for larger models.**
>
> The evasion loss is very computationally efficient, requiring only an L2 distance computation for each parameter vector. This is comparable to the compute required for L2 regularization.
>
> The evasion loss does incur a memory overhead for loading the model, since it requires loading an "anchor model" fixed to the initialization. For the ResNet-50 scale models that we use (described in Section 5), this is inconsequential. However, for LLM-scale models it could present some moderate challenges, although it would certainly still be a viable attack.
>
> **Reason for using the selected datasets.**
>
> Most neural trojan papers only train a handful of models, or a few hundred at most. Investigating model-level detection requires training datasets of models. We train over 6,000 models in total, with each individual number in our tables representing hundreds of trained models. Each model requires approximately an hour to train (more for Vision Transformers on GTSRB). Thus, for computational reasons we decided to focus on 32x32 datasets that are commonly used in the model-level detection literature.
>
> **Differences between Figure 3, Table 2, and Table 1.**
>
> You are correct that Figure 3 and Table 2 show two views of the same data. Table 2 is a table of AUROC values, and Figure 3 shows individual ROC curves for three representative experiments in this table. It's standard to show ROC curves alongside AUROC values, as the full curves can provide helpful context as to the shape of the tradeoff and whether one method is always better than the other, or whether this is only true for certain recall and FPR values.
>
> Table 1 shows different numbers: the ASR of the trojans and accuracy of the trojaned models on clean inputs without triggers. This demonstrates that our evasive trojans have similar ASR and accuracy to standard trojans and clean networks, which is important for ensuring that the comparisons between standard and evasive trojans are fair. If we have addressed the thrust of your concerns, we kindly ask that you consider raising your score.
>
> **Reason for Param detector's high performance.**
>
> In Appendix B.1, we show that removing our randomization loss causes the Param detector to obtain 100% AUROC. In fact, the Param detector's strong performance in preliminary experiments is what led us to develop the randomization loss and discover the issue of emergent coordination that we show in Figure 4. Even after including our randomization loss, the Param detector performs better on our Evasive Trojans than on Standard Trojans, although the difference is much smaller than before. We have updated the paper to clarify this. Thank you for your suggestion.

---

### Meta-Review · Area_Chair_gNhE · 2023-12-11

**Metareview:**

The authors suggest a new way to inject trojans into neural networks which are hard to detect based on model-level detection. The trojan network is trained with three particluar loss components: a distribution matching loss ensures that logits and parameters of the clean and trojan networks are similar, a specificity loss ensures high specificity for the trigger and a randomization loss makes sure that certain summary statistics of clean and trojan models are aligned with a random vector.

Strengths:
- Paper is well written and easy to understand
- evasion of several baseline detectors

Weaknesses:
- lack of comparison to existing baselines (some of them could be resolved in the rebuttal)
- only low resolution datasets
- non-consistent losses: for MNIST a different randomization loss is used (see Appendix A)
- the hyperparameter choice (Appendix A) is unclear and the training details regarding use of dropout seem highly specific to the employed networks

**Justification For Why Not Higher Score:**

see above

**Justification For Why Not Lower Score:**

N/A

---

### Decision · Program_Chairs · 2024-01-16

Reject